# Designing Calcium Phosphate Nanoparticles with the Co-Precipitation Technique to Improve Phosphorous Availability in Broiler Chicks

**DOI:** 10.3390/ani11102773

**Published:** 2021-09-23

**Authors:** Diana A. Gutiérrez-Arenas, Manuel Cuca-García, Miguel A. Méndez-Rojas, Arturo Pro-Martínez, Carlos M. Becerril-Pérez, Maria Eugenia Mendoza-Álvarez, Fidel Ávila-Ramos, Jacinto Efrén Ramírez-Bribiesca

**Affiliations:** 1Departamento de Medicina Veterinaria y Zootecnia, División de Ciencias de la Vida, Campus Irapuato-Salamanca, Universidad de Guanajuato, Ex Hacienda El Copal Km. 9, Carretera Irapuato-Silao, A.P. 311, Irapuato CP 36824, Gto., Mexico; diana.gutierrez@ugto.mx (D.A.G.-A.); ledifar@ugto.mx (F.Á.-R.); 2Programa de Ganadería, Instituto de Recursos Genéticos y Productividad, Campus Montecillo, Colegio de Postgraduados, Carretera Mexico-Texcoco Km. 36.5, Montecillo, Texcoco CP 56230, Méx., Mexico; jmcuca@colpos.mx (M.C.-G.); aproma@colpos.mx (A.P.-M.); color@colpos.mx (C.M.B.-P.); 3Departamento de Ciencias Químico-Biológicas, Universidad de las Américas Puebla, Ex Hacienda Santa Catarina Mártir S/N, San Andrés Cholula CP 72810, Pue., Mexico; miguela.mendez@udlap.mx; 4Instituto de Física, Benemérita Universidad Autónoma de Puebla, Av. San Claudio, Col. San Manuel, Puebla CP 72570, Pue., Mexico; emendoza@ifuap.buap.mx

**Keywords:** nanotechnology, calcium diphosphate, brushite, poultry, nutrition

## Abstract

**Simple Summary:**

The increase in phosphate prices has inflated the cost for poultry feeding. Dicalcium phosphate is an essential mineral involved in the metabolism and development and is commonly used as a dietary source of phosphorus (PT) for poultry. The use of nanoparticles of dicalcium phosphate (NDP) could increase the bioavailability of PT in the diet. The sizes of the nanoparticles formed were 20 and 80 nm. NDP had the Ca:P ratio 1:1.12. The digestibility of PT in birds improved by 67% in the treatment with 0.35% available P (Pa) of NDP. The highest contents of PT -breast were found with the levels of 0.35 and 0.46% Pa of NDP. In conclusion, the use of NDP as an ingredient for broilers was efficient with a Pa dose at 0.35%. This dose was ideal in chicks for digestibility and absorption values. Additionally, results showed an improvement in the amount of PT in breast.

**Abstract:**

Dicalcium phosphate (DP) is a mineral involved in the metabolism and development and is used as a dietary source of phosphorus (P_T_) for poultry. Our study objective is to design nano-dicalcium phosphate (NDP) by co-precipitation. The methodological procedure was divided into two phases: (1) NDP synthesis, and (2) bird performance, digestibility, and Ca-P in chick’s tissues. The sizes of the NDP were 20–80 nm. NDP had the Ca: P ratio of 1:1.12. The birds were divided into control diet (available P (P_a_) = 0.13%) and three supplementary P sources [Commercial (Calcium phosphate), analytical grade (DP) and nanoparticles (NDP)] with three P_a_ levels (0.24, 0.35, 0.46%). Supplementary P sources compared to the control treatment had the highest body weight gain (698.56 vs. 228; *p* < 0.05) and feed intake (FI) (965.18 vs. 345.82), respectively. The digestibility of P_T_ (67%) improved with 0.35% NDP. The highest contents of P_T_ -breast were with the levels of 0.35 and 0.46% NDP. The P_T_, ash, and diameters were higher (*p* < 0.05) with supplementary P compared to the control treatment. As conclusion, the use of 0.35% NDP was the ideal dose in the chicks for the digestibility, absorption values, and the amount of P_T_ in the breast.

## 1. Introduction

Phosphorus (P) is an essential nutrient involved in various metabolic processes, including calcium (Ca) metabolism. Both minerals are closely associated; thus, the deficiency or excess of either interferes with the other’s metabolism. P requirement for broilers is high cost with little availability, and even though cereals used in chicken feed have phytic acid molecules (phosphorus ~75%), their digestive tract cannot break down this compound for obtaining P [1]. As broilers rapidly reach a weight of ~2.6 kg, Ca and P deficiencies become more common and can induce rickets and tibial dyschondroplasia, causing lameness and high mortality rates [2]. Macromineral imbalances increase the ingested P excretion. Resulting in high P concentrations accumulated in soils, eventually polluting groundwaters by eutrophication [3].

The first recommended concentrations of Ca and P for growing chickens were published by the NRC in 1984 [4], and suggest a maximum ratio of 2.28 Ca to 1 P. Although, P availability and absorption depend on the food source and the physiological digestive processes of the birds. Specifically, P absorption occurs through the intestinal brush border membrane, involving Na-dependent and Na-independent pathways [5]. The most important Na-P cotransporters are type I and III (PiT1 and PiT2), both expressed mainly in the intestinal membrane of the brush border. PiT2 appears to play an important role in the regulation of intestinal P absorption [5,6]. Although, these cotransporters have not yet been well studied in chickens, the duodenum of birds has the highest level of cotransporters [6], giving better P absorption efficiency than jejunum or ileum [1]. Additionally, vitamin D in chicks also improves P permeability in the intestine and participates in the transfer of P in all small intestine segments [1,2,3,4].

Poultry diets are supplemented with food-grade inorganic sources of phosphorus, such as monocalcium, dicalcium, sodium, magnesium phosphates, and defluorinated tricalcium phosphate. In particular, dicalcium phosphate (DP: CaHPO_4_) has relevance in several biological, nutritional, and industrial applications. It is an essential mineral involved in the metabolism and development of farm animals, a common dietary source of phosphorus (P) for poultry [2]. The increase in phosphate prices has inflated poultry feeding costs. Many research groups worldwide are interested in developing synthetic methods to prepare nanostructured calcium phosphate, which would allow for the control of their chemical and physical properties like their size, shape, biocompatibility solubility, and more [7]. The use of DP nanoparticles could increase the bioavailability of P in the diet. A smaller particle size would result in a larger exposed surface for chemical interaction inside poultry gastrointestinal tube, yielding a higher absorption efficiency [7,8]. Sedaghat et al. [9] reported that most particles measuring up to 100 nm are often absorbed in the intestinal tube 15 to 250 times more than those of bigger nanoparticles size, thus, reducing the amount of DP needed, resulting in lower feeding costs.

There are several methods available for the preparation of DP nanoparticles. Some of the most popular are sol-gel, hydrothermal synthesis, mechano-chemical preparation, milling of natural bones, among others [9,10]. For the preparation of nano-dicalcium phosphate (NDP), the use of different chemical precursors for calcium: Ca [CaCl_2_, Ca(NO_3_)_2_, CaCO_3_, Ca(CH_3_COO)_2_] or P [(NH_4_)2HPO_4_, NH_4_H_2_PO_4_, KHPO_4_, N_2_HPO_4_, NaH_2_PO_4_, Na_2_HPO_4_] have been explored [8,9,10]. Additionally, the nature of the chemical precursors will affect the characteristics of the final product, as well as the proper selection of surface modifiers (surfactant). Some of the most explored surface modifiers are polyacrylic acid, citric acid, several amino acids, ethylenediamine tetra-acetic acid. The objective of this study was focused on the design of NDP by chemical co-precipitation, a simple and easily adaptable method for large-scale production that may be useful for the preparation of high-quality feedstock products for the poultry feed industry.

## 2. Materials and Methods

The methodological procedure was divided into two phases:

**Phase 1**: Synthesis and characterizing of calcium phosphate nanoparticles: NDP were made by the co-precipitation method using reagents available in Ca and P. NDP were characterized by scanning electron microscopy (SEM), transmission electron microscopy (TEM), dynamic light scattering (DLS), and zeta potential measurements. The chemical composition of the NDP was analyzed with SEM integrated an energy-dispersive X-ray spectroscopy (EDS) and powder X-ray diffraction (pDRX).

**Phase 2**: Bird performance, digestibility, and Ca-P content in tissues: Broiler chicken treatments (NDP included) were evaluated in weight gain, feed intake, feed conversion, In vivo digestibility trail, and Ca-P content in tissues.

### 2.1. Phase 1: Synthesis and Characterizing of Calcium Phosphate Nanoparticles

#### 2.1.1. Reagents and Solutions

Anhydrous dibasic sodium phosphate (Na_2_HPO_4_) brand J. T. Baker. PM 141.96 (Solution at 10.8 mM), anhydrous calcium chloride (CaCl_2_) brand JT Baker PM 110.99 (Solution at 18 mM), polyvinylpyrrolidone PV40T-500G brand Sigma-Aldrich^®^ (Naucalpan, Mexico) (1% solution), hydrochloric acid (HCl) brand JT Baker. P.M. 36.46 (20% solution), distilled water. Anhydrous dibasic sodium phosphate (Na_2_HPO_4_) (10.8 mM). 50 mL Na_2_HPO_4_ × (10.8 mM/1000 mL) × (1 M/1000 mM) × (141.96 g/1 Mol) = 0.07666 g. Anhydrous calcium chloride (CaCl_2_) (18 mM). 50 mL CaCl_2_ × (18 mM/1000 mL) × (1 M/1000 mM) × 110.98 g/1 Mol) = 0.09988 g. Hydrochloric Acid (HCl) (20%). 20 mL of concentrated HCl was added in 80 mL of distilled water. Polyvinylpyrrolidone (PVP) PV40T-500G (Sigma-Aldrich^®^) (1%). 0.5 g was added in 50 mL of distilled water. The NDP was synthesized according to the following chemical reaction: Na_2_HPO_4_ + CaCl_2_ → CaHPO_4_ + 2NaCl.

#### 2.1.2. Manufacture of Nanoparticles

Sodium phosphate (Na_2_HPO_4_) and calcium chloride (CaCl_2_) were separately dissolved in water and calibrated at 108 mM and 180 mM, respectively. The synthesis of Nano was carried out with the co-precipitation method; initially the Na_2_HPO_4_ solution was added dropwise to the CaCl_2_ solution, keeping the mixture under constant stirring, at room temperature and pH 5–6. Subsequently, the prepared solution was kept in constant stirring. The PVP polymer was added dropwise, avoiding agglomeration of the nanoparticles formed. The last phase was the washing of the nanoparticles, which was done with duplicate centrifugation (two cycles of 5 min at 8000 rpm) to remove the salt (NaCl) formed during the reaction. The purification of the nanoparticles (powder) was obtained by lyophilization at −46 °C. Figure 1 summarizes the design of the nanoparticles.

#### 2.1.3. Characterization of the Calcium Phosphate Nanoparticles

##### Scanning Electron Microscopy

The samples were analyzed by SEM with the JEOL, JSM 35-C to obtain topological and morphological data. The beam landing energy cam lowered was 30 KeV at 50 ev and a resolution of 1.4 nm at 1 nm at 15 kV. The samples were covered with gold before performing the SEM analysis, and pDRX was performed with SEM analysis.

##### Transmission Electron Microscopy

The TEM was used to characterize the shape and size of the nanoparticles. The electron microscope used was a JEM 1010 60kV brand JEOL IPN, Mexico. The samples for the TEM measurement were prepared with the addition of a drop in colloid solution, on a copper grid (400 mesh) and covered with an amorphous carbon film, inducing the evaporation of the solvent in air at room temperature.

##### Dynamic Lightscattering and Zeta Potential Measurements

Samples of 0.005 g of NDP dispersed in 10 mL of liquid suspension were measured for particle size by DLS and zeta potential using a Zetasizer Nano ZS equipment (Malvern Nano-ZS (Worcestershire, UK) laser: λ = 532 nm) at 25 °C. A 654 nm He–Ne laser was used as the light source and an avalanche photodiode detector. The particle size was measured by DLS with a refractive index of 1.5394 and absorbance of 0.01, using an immersion cell (zen1002, Malvern Instruments) with a pair of parallel Pd electrodes to cause an electrical trigger on charged particles. The recorded signals were analyzed with the Zeta-sizer software, and the sample size for the surface zeta potential measurement was limited to measurements of 4 mm (length) × 7 mm (width) × 1.5 mm (height).

##### Chemical Composition

The chemical composition of the NDP was analyzed with SEM (Physics Institute, BUAP, Puebla, Mexico), integrating an energy-dispersive X-ray spectroscopy (EDS), Inca-X-act pentaFET Oxford Instruments (Oxford, UK). The NDP samples were analyzed with an infrared spectrophotometer, Fourier Varian Scimitar FTIR800 (New York, NY, USA), generating vibrational spectra necessary to analyze and compare the data obtained with original standard samples (DP analytical grade. Baker) (Almelo, The Netherlands). The spectra were obtained in solid state, using an attenuated total reflectance detector (Almelo, The Netherlands), with a germanium glass window and recording the data in the region between 4000 and 400 cm^−1^. The crystalline phase of NDP was made by pDRXD using a Bruker-AXS D5000 diffractometer (Physics Institute, BUAP-Mexico, Puebla, Mexico). The samples were finely pulverized and deposited in a quartz sample holder, using the K line of an emissive copper source (=1.5418 Å) in mode 2, with measuring intervals of 10–80° and scanning steps of 0.02° and time intervals of 0.6 s.

### 2.2. Phase 2. Bird Performance, Digestibility, and Ca-P Content in Tissues

This study was conducted at the Poultry Research Unit of Postgraduated College, Montecillo-Texcoco, Mexico. All procedures involved in the experiment were regulated by the standards of ethics and animal welfare required by the Official Mexican Standard published under the Mexican Council on Animal Care guidelines [11]. The trail infrastructure complies with the operation regulated by the Official Mexican Standard [12]. The sacrifice of the birds was carried out with the requirements of the Official Mexican Standard [13].

#### 2.2.1. Animals Management and Diets

One-hundred-day-old male Ross chicks vaccinated against Marek and Newcastle were selected for the study. The birds were weighed and distributed randomly in ten treatments, with five repetitions of two chicks each. All birds were housed in pairs in electric brooders in battery with temperature control, well ventilated, with artificial light (white fluorescent), water and feed were supplied ad libitum. The birds had good suitable conditions during the experimental period of 21 d. Each replica of chicks was housed in a space of 0.40 × 1.20 m^2^, maintained daily with continuous light until the end of the first 7 days of age, then they received 22 h of light per day. The animals were inspected twice a day to record hygienic conditions, health, feeding, and number of dead birds. Control diet (only ingredients: available P (P_a_) = 0.13%) and three supplemental P sources (Commercial (Calcium phosphate), analytical grade (containing DP) and nanoparticles (NDP)) with three P_a_ levels (0.24, 0.35, and 0.46%) were evaluated in a 4 × 10 factorial arrangement of treatments. The commercial source is commonly used in the poultry industry, the design of the NDP is the treatment of interest, and the analytical group was included as a positive control, knowing that it is highly available in P. During the preparation, the aliquot of NDP (20 g) was mixed with soybean meal (280 g) and stirred for 20 min with a room temperature plate shaker. Subsequently, the compound mixture was added with the other ingredients of the diets, calculating the doses of supplemented P for each treatment. Rations were formulated to meet nutritional requirements for growing broiler chicks [14,15] except for Ca and P, as indicated in Table 1.

#### 2.2.2. Bird Performance

Chicks were weighed (electronic scale; Torrey Til/s: 107 2691, Houston, TX, USA) from day 0 until 21 d of the trial and the average body weight gain (BWG) per treatment was calculated. Diets were prepared weekly and stored in plywood boxes. The total feed was offered in one meal, in the morning, and the feed refusal (target was 5% orts) was collected before each morning. The amount of feed offered to each animal group was adjusted in dry matter (DM) basis per day to get the total feed intake (FI) and feed conversion ratio (FC) over the period of 21 d.

#### 2.2.3. Digestibility Trial

Six birds by treatment were randomly selected to conduct nutrient digestibility trial. Total droppings were collected in a container (1 cm × 35 cm × 70 cm) with a grid for each replicate during two periods within the experiment: a) 7 to 10 d, and b) 17 to 20 d. All excreta were stored in a freezer (−20 °C) until analysis. Phosphorous was calculated with the following formula:

Apparent P digestibility (A_DP_, %) = [(Total P intake (T_PI_) – total P excreted (T_PE_)/T_PI_)] × 100.

Total P absorbed (T_PA_) = T_PI_ × A_DP._

#### 2.2.4. Tissues Collection and Laboratory Analysis

At the end of the trial, frozen samples of experimental diets and feces were thawed overnight at room temperature and analyzed for DM by drying in an oven at 65 °C for 48 h (AOAC, 2016). Dried food and feces samples were ground through a 1-mm screen using a Christy-Norris mill (Christy and Norris Ltd., Chelmsford, UK). All animals were sacrificed by cervical dislocation following the corresponding national ethical guidelines cited above. Breast and liver samples were extracted and kept frozen at −20 °C. Tibia bones were cleaned in boiling water and then the fat extracted using ethyl ether for 6 h and dried in an oven at 105 °C for 1 h [16]. The ash content [17] was obtained from the left tibia bones (L_TB_), and measurements were made on the bones of the right tibia (using a Vernier caliper -Truper) and classified into: R_TB_T_L_ = Total length of the right tibia; R_TB_T_PD_ = Proximal diameter of the right tibia; R_TB_T_MD_ = Medial diameter of the right tibia; R_TB_T_DD_ = Distal diameter of the right tibia.

Samples were subjected to all or part of the following analysis: DM and ash (600 °C in an oven) [17]. Diets, excreta, and tissues samples were digested in nitric perchloric and fluorhydric acid mixture and subsequently concentration of P was analyzed using a Cary IE UV-Vis spectrometer, following the methods indicated by AOAC [17]. Phosphorous release from NDP was measured by suspending them in phosphate buffer saline (pH 7.4) at 1 mg mL^−1^ of P based on entrapment efficiency. Samples were divided into 5-mL aliquots and transferred to a shaker under constant agitation at 110 rpm and 29 °C. Samples were filtered using 0.2-μm cellulose membrane syringe (Millipore Sigma, Naucalpan, Mexico) and then P content was also analyzed.

### 2.3. Statistical Analysis

Phase 1 only describes the values obtained in the design and characterization of the NDP. Phase 2: Data were analyzed as a completely randomized design using the MIXED procedure of SAS software [18]. The pen of the chicks served as the experimental unit, and the main factors used in the model were P level (Control, one level; Supplementary P: 3 levels), three sources and their interaction were also included. Dietary treatments and the source of P were taking into account, linear and quadratic effects were performed within the treatments. The results are reported as least square means and standard error. Significant differences among treatment means were determined by Tukey’s multiple range test with a *p* ≤ 0.05 level of probability.

## 3. Results

### 3.1. Phase 1: Synthesis and Characterizing of Phosphate Dicalcium Nanoparticles

#### 3.1.1. Characterization

The characterization of the NDP was done. The crystalline particles of dicalcium phosphate stabilized superficially with PVP, and a prismatic shape (Figure 2a) was observed by SEM in both photos; the enlarged image shows the morphology of the crystals. The sizes of the nanoparticles formed and measured by TEM were 20 and 80 nm (Figure 2b). However, the average size of the NDPs through the Nanotrac Wave registered populations with an average size of 23.5 nm to 2477 µm, with a bimodal size distribution (Figure 2c). The majority population had a value of approximately 26 nm. The data from the zeta sizer confirm the presence of agglomerates with measurements of 141 nm diameter (Figure 2d).

#### 3.1.2. Composition

Figure 3a shows the results obtained by DLS. Two zones were exhibited in the Ca: P stoichiometry, showing homogeneity in the analyzed samples. The photos indicated the Ca: P ratio of 1:1.11 and 1:1.12. The FTIR analysis by EDS showed the bands associated with P–O vibrations, suggestive of the presence of phosphate groups. The 3-band symmetry between 980 and 1120 cm^−1^ (Figure 3b) is indicative of brushite (CaHPO_4_ • 2H_2_O) or monetite (CaHPO_4_) formation. The pDRX demonstrated a stoichiometric reaction between dibasic sodium phosphate and calcium chloride (Figure 3c), indicating a higher quantity of calcium phosphates in the monetite phase with some hydroxyapatite residues.

### 3.2. Phase 2. Bird Performance, Digestibility, and Ca-P Content in Tissues

#### Bird Performance

Table 2 shows the bird performance results. Supplementary P sources compared to the control treatment had the highest BWG (698.56 vs. 228; *p* < 0.05) and FI (965.18 vs. 345.82), respectively. The three sources of supplemental P showed a linear effect in BWG (0.24%: 603.10 vs. 0.46%: 779.4; *p* < 0.05), and a quadratic effect was presented in FI (*p* < 0.05). Particularly the P levels with 0.35 and 0.46 (~FI: 1032 g) were higher than the diet with 0.24% P (FI: 830 g). The FC did not show significant differences (*p* > 0.05) between the treatments with supplementary P. In general, the global values in growth performance did not show significant differences (*p* > 0.05) with the three sources of supplementary P.

Table 3 shows the results of the digestibility and apparent absorption of P in the chicks during the 2 measurement periods. T_PI_ of the control groups was lower in both measurement periods (*p* < 0.05) than the groups with supplementary P. During the first period (10 d) T_PI_, T_PE,_ and T_PA_ showed a linear increase (*p* < 0.05) according to the levels of supplemented P. However, the digestibility of P (67%) improved in the treatment with 0.35% of NDP. The values evaluated at 21 d, also presented a linear effect (*p* < 0.05) in the variables T_PI_ and T_PE_ of the commercial and NDP treatments, while analytical group presented a quadratic effect in T_PI_ and T_PE_. The highest digestibility indices were presented in the groups with 0.24 and 0.35% of commercial, 0.24% of NDP, and 0.24 and 0.46% of analytical groups. On the other hand, the consumption of P was higher (*p* < 0.05) at 21 d than 10 d. The averages of the A_DP_ for the 10 d did not have significant differences (*p* > 0.05) among the supplementary P levels: C = 58.4%, commercial = 63.1%, NDP = 63.5%, and analytical = 55.5%. The mean digestibility’s of P increased at 21 d for: C = 64.1%, NDP = 71.5%, and analytical = 61.4%. 

The P content and measurement of some tissues is shown in Table 4. Breast P_T_ was higher with supplementary P sources vs. P_T_ -control (mean 331 vs. 290, *p* < 0.05). In particular, the highest contents of P_T_ -breast were found with the levels of 0.35 and 0.46% P of NDP compared to the P_T_ commercial and analytical treatments (mean 395 vs. 313, *p* < 0.05). The P_T_ -liver was similar (*p*> 0.05) between the P_T_ -control and P_T_ of 0.35% P of commercial—analytical and 0.46% P of NDP. Meanwhile, the highest contents in P_T_ -liver were at the levels of 0.24% P commercial—NDP, 0.24 and 0.46% P in analytical treatments compared to the P_T_ of the other treatments (mean 498 vs. 382, *p* < 0.05). The P_T_ -tibia bone had the highest content with levels of 0.35% P in commercial—analytical treatments, and 0.46% P in commercial—NDP compared to the P_T_ -control (mean 19.092 vs. 14.980; *p* <0.05). The ash content was higher in the three treatments with supplementary P vs. control treatment (mean 48.93 vs. 34.52, *p* < 0.05). The highest ash contents were with 0.35 and 0.46% P levels vs. 0.24% P of the three treatments (mean 51.02 vs. 44.76, *p* < 0.05). L_TB_ was higher in the three treatments with supplementary P compared to the L_TB_ -control (66.96 vs. 52.27, *p* < 0.05). The most lengths in L_TB_ occurred at the levels of 0.35 and 0.46% P compared to the 0.24% P of the three treatments (mean 69.31 vs. 62.24, *p* <0.05). The diameters evaluated in the tibia bones with supplementary P were greater than the control treatment: R_TB_T_PD_ (17.46 vs. 12.02, *p* < 0.05), R_TB_T_MD_ (6.03 vs. 4.27, *p* < 0.05), R_TB_T_DD_ (14.82 vs. 10.82, *p* < 0.05). While the three diameters of the tibia evaluated with the sources of P, did not show relevant significant differences (*p* > 0.05).

## 4. Discussion

### 4.1. Phase 1: Synthesis and Characterizing of Calcium Phosphate Nanoparticles

Phosphorous is the third most expensive component of the poultry diet after energy and protein, so alternatives to improve the bioavailability of P is in our best interest. Usually the sources of dicalcium phosphates, obtained from phosphate rock, are the most used in the feeding of farm birds [19]. The alternatives of finding P supplements with a high biological value can improve availability, reduce the supplement cost, and reduce contamination in the environment. Some studies carried out with mineral nanoparticles indicate that the availability and absorption at the intestinal level are improved, such as in the case of selenium in ruminants [20]. In the case of the design of Ca-P nanoparticles, there is extensive information indicating its approach to the remineralization of bone and dental tissues, and controlled drug release [5,6,7,8,9,10]. However, the high purity and safety of nanoparticles used in humans are expensive when used in farm animals. The NDPs designed in this study focuses on increasing P availability in the intestinal tract, for which a hygiene protocol was followed but without sterilization. The co-precipitation technique is a very simple technique (described above); it is commonly used to separate and concentrate oligoelments from a solution, regulating the pH and solubility of the mineral precipitate. In this case, no published data indicate a co-precipitation technique to design P: Ca nanoparticles. Other techniques reported are with wet chemical, sol-gel, mechanochemical, solid-state, ultrasonic field, and electric discharge [21,22,23,24,25,26]. The co-precipitation method is suitable for simple operation, low production cost, and the possibility to scale the design in large quantities. In this study, the production efficiency with this technique was 94%. The NDPs had a uniform particle size range of 20 to 80 nm. The NDPs size depends on the technique, the polymer or surfactant, and the encapsulation process. For example, with the ball milling technique, sizes are cited from 46.6 to 62.6 nm [27]. Associated with the polymer, the nanoparticles designed with polyacrylic acid had a 5–60 nm [28]. The NDPs manufactured with the spray drying technique had an approximate size of 116 nm [29]. In another similar study in chickens, the authors manufactured NDPs using the sol-gel technique with satisfactory results [21]. However, the article does not describe the procedure in detail. The sol-gel technique is widely disseminated in the design of metallic nanoparticles, from the use of inorganic salts with a calcination process at maximum temperatures of 700 °C. The routine use of the technique is mainly to form oxides and microbicidal substances with industrial and medical applications [30]. Referring to the NDPs characterization, the Ca:P stereochemistry was homogeneous and ideal with a maximum Ca: P ratio of 1: 1.12, identifying dibasic sodium phosphate and Ca chloride groups with residual amounts hydroxyapatite. Calcium phosphate is a highly biocompatible biomaterial [26]; the mineralized form known as hydroxyapatite is one of the primary inorganic components. The various forms of CaP include amorphous Ca phosphate, initially formed from an aqueous solution supersaturated with Ca and phosphate cations. NDPs synthesis from Na phosphate and Ca chloride precursors was carried out in room temperature, under constant stirring and controlled pH. The NDPs formed in this study had the ideal physicochemical characteristics to be used as a source of P in poultry feeding, as reported in another similar study where NDPs synthesis was carried out with the ball mill technique [26].

### 4.2. Phase 2. Bird Performance, Digestibility, and Ca-P Content in Tissues

The growth performance in broilers depends on nutritional requirements, zootechnical management, welfare, and environmental conditions. Supplementary P in the diet also influences the effective response of farm birds [16]. In this study, all supplementary P sources compared to the control group improved BWG and FI. Mainly, P participates in skeletal development and energy metabolism, while Ca is mainly associated with the structure of the skeleton [31]; thus, the deficiency of both minerals affects the biological development of tissues in animals. The contribution of P in the chicks’ diets is mainly from cereal grains and by-products; however, the amount provided by the ingredients is minimal and supplementary sources of P are required to cover the requirement [31]. For example, the BWG in chicks aged 18 and 28d was improved with levels of mono and dicalcium phosphates at doses of 0.25 to 0.35% [16,17,18,19,20,21,22,23,24,25,26,27,28,29,30,31,32]. In our study, the three sources of supplementary P had a linear effect on BWG (0.24 vs. 0.46%), considering the levels of 0.35 and 0.46% P as the best value [33]. Specifically, Ca phosphates have high availability, however, the sources and levels of P in the diets are expensive. As already mentioned, several studies indicate that NDP improve the absorption and bioavailability of P; and mainly the best availability is given by particle size, shape, porosity, surface area, and surface load [26]. In this study, the use of the NDP showed productive benefits with the levels of 0.35 and 0.46%, the most important factor was the size of the particle and its stability, as previously mentioned, the NDP measured 80 nm, facilitating the absorption of the P [29,30,31,32,33,34]. Contrarily, Ca absorption is better when the particle size of 137–388 µm increased the weight gain, while the very small (28 µm) or very large (1306 µm) particles did not improve weight gain [35]. However, the Ca and P absorption process is not simple, various physiological events act at the intestinal level, highlighting the action of the intestinal alkaline phosphatase enzyme [36], parathyroid hormone, calcitonin, vitamin K, and vitamin D [37]. In this case, the data observed with the NDP were constant, there was good bioavailability of P, and the data between the NDP and positive control (analytical) groups were similar.

The biological availability of P was measured with body weight gain, feeding efficiency, and bone mineralization in the tibia and femur [38,39]. The most utilized P sources in the poultry industry have been di and tricalcium phosphate, sodium and potassium phosphates. Several authors cite that the bioavailability of P at doses of 0.45 to 0.73% improves phosphate fixation in the bone to form hydroxyapatite crystals [40]. Thus, the amount of ash in the tibia determines bone mineralization [41]. Several methodologies evaluate bone mineralization, including ash content, resistance to breakage, ultrasound, and the direct determination of Ca and P in long bones [42]. Various authors have reported chicks of 22 d femur and tibia lengths of 49 to 54 mm [43] and 67.2 mm [44], respectively.

In our study, the P_T_ L_TB_ and ash tibia bone were higher with the supplemented P levels. The length of the bones is closely related to the weight of the bird and the sex; male-birds have longer leg bones than females [45]. The Ca: P ratio in our study was 1.12, but other studies indicate better bone mineralization with a ratio greater than 1.39 [27]. The ash content of this study ranged from 44 to 53%, thus considered within normal values in birds [46]. However, other authors [47] mention that broiler chicks can be adapted to low Ca and P levels in the diet, concluding that P maximizes the best effective response and good mineralization of the tibia is obtained. As is known, the P absorbed by the intestinal route is rapidly transferred to the skeletal and soft tissues. In our study, the doses of 0.35 and 0.46% of P supplemented with NDP improved the content of P in breast (390–400 mg P/100 mg of meat). The suggested P requirement for humans by the recommended dietary allowance is 700 to 1250 mg/day [48], so chicken meat is considered an important source of highly available P mineral needs in humans. The NDP design was 30% more expensive than the commercial phosphorous source. However, the use of NDPs in the poultry industry is justified by the benefits already noted above.

## 5. Conclusions

The data confirmed that NDP contains a chemical formula of CaHPO_4_. The co-precipitation technique showed crystalline monodisperse prismatic nanoparticles (brushite and hydroxyapatite phases), with diameters between 20 and 80 nm. Calcium phosphates in nanometric form were amorphous with Ca/P ratios from 1.0 to 1.12. The NDPs were easily dispersible in water after PVP coating, showing good stability. NDPs were highly soluble and the small particle size and surface area made them potentially useful for intestinal absorption. The evaluation of NDP as a source of nutritional P for broilers showed no adverse effects, and mortality was within the standard range for common sources of P in diets. The NDP did not show significant changes in the performance of the birds, compared to the commercial grade P sources and analytical P source (positive control). Manufacturing NDP was 30% more expensive than commercial P, but its use of NDP as an ingredient for broilers was efficient with the dose of P supplemented at 0.35%. This dose was the ideal one in the chicks for the digestibility and absorption values; the amount of P in the breast improved. Further research is needed on the biodistribution of NDP in tissues and its long-term effect on physiological development and productive stages in farm birds.

## Figures and Tables

**Figure 1 animals-11-02773-f001:**
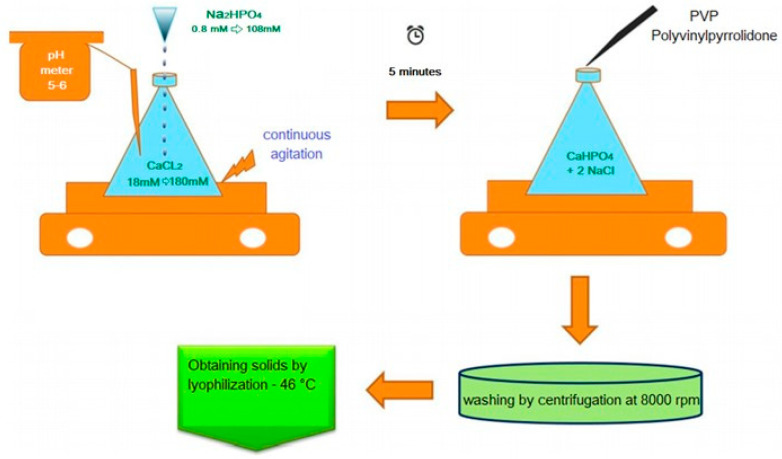
Design procedure for calcium phosphate nanoparticles.

**Figure 2 animals-11-02773-f002:**
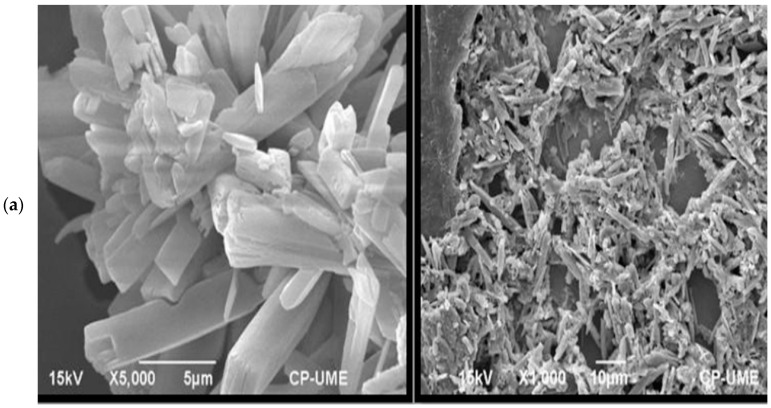
Characterization of phosphate dicalcium nanoparticles. Scanning electron microscopy integrated with energy-dispersive X-ray spectroscopy and powder X-ray diffraction. (**a**) Prismatic form of phosphate dicalcium nanoparticles, (**b**) Transmission electron microscopy micrographs of phosphate dicalcium nanoparticles, (**c**) Size measured with dynamic light scattering on Zeta-sizer in phosphate dicalcium nanoparticles, (**d**) Size distribution by means of dynamic light scattering in phosphate dicalcium nanoparticles.

**Figure 3 animals-11-02773-f003:**
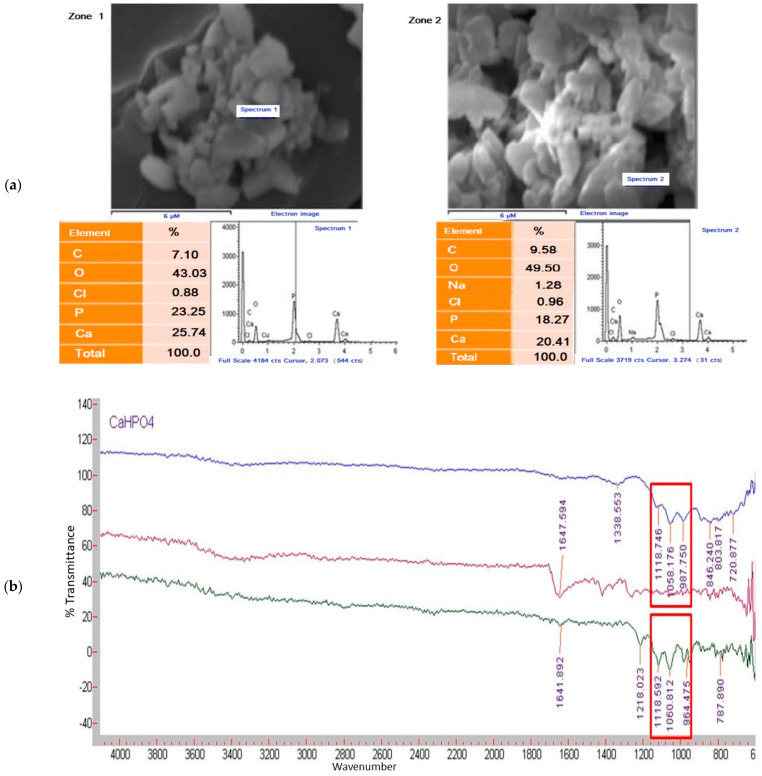
Chemical composition of phosphate dicalcium nanoparticles. (**a**) Analysis of phosphate dicalcium nanoparticles by energy-dispersive X-ray spectroscopy, (**b**) Comparison of nanoparticles of phosphate dicalcium nanoparticles synthesized (bottom), with polyvinylpyrrolidone (middle) and CaHPO_4_ of analytical grade (top), (**c**) X-ray diffractograms: *i)* calcium phosphate nanoparticles; *ii)* brushite- hydroxyapatite.

**Table 1 animals-11-02773-t001:** Ingredient composition of experimental diets in chicks fed with three sources and levels of supplemental phosphorus.

	Control	Commercial			NDP			Analytical		
P_a_ %	0.13	0.24	0.35	0.46	0.24	0.35	0.46	0.24	0.35	0.46
Ingredients										
Sorghum	58.64	58.21	57.74	57.31	58.22	57.81	57.38	58.48	58.30	58.14
Soybean meal	31.90	31.99	32.09	32.18	31.99	32.07	32.16	31.93	31.97	32.00
Soybean oil	5.35	5.48	5.62	5.75	5.48	5.60	5.73	5.40	5.45	5.50
CaCO_3_ ^1^	2.52	2.21	1.91	1.60	2.11	1.70	1.30	2.10	1.69	1.27
Ca orthophosphate ^2^	0.00	0.52	1.05	1.57	0.00	0.00	0.00	0.00	0.00	0.00
NDP ^3^	0.00	0.00	0.00	0.00	0.61	1.22	1.84	0.00	0.00	0.00
Analytical CaHPO_4_ ^4^	0.00	0.00	0.00	0.00	0.00	0.00	0.00	0.50	1.00	1.50
L-Lysine	0.35	0.35	0.35	0.35	0.35	0.35	0.35	0.35	0.35	0.35
DL-Methionine	0.43	0.43	0.43	0.43	0.43	0.43	0.43	0.43	0.43	0.43
L-Threonine	0.14	0.14	0.14	0.14	0.14	0.14	0.14	0.14	0.14	0.14
L-Triptophane	0.02	0.02	0.02	0.02	0.02	0.02	0.02	0.02	0.02	0.02
Vit and mineral mix ^5^	0.30	0.30	0.30	0.30	0.30	0.30	0.30	0.30	0.30	0.30
NaCl	0.35	0.35	0.35	0.35	0.35	0.35	0.35	0.35	0.35	0.35
Nutritional intake calculated										
ME (Mcal kg^−1^)	3200	3200	3200	3200	3200	3200	3200	3200	3200	3200
Crude protein, %	22.00	22.00	22.00	22.00	22.00	22.00	22.00	22.00	22.00	22.00
Lysine, %	1.25	1.25	1.25	1.25	1.25	1.25	1.25	1.25	1.25	1.25
Methionine + Cystine, %	0.94	0.94	0.94	0.94	0.94	0.94	0.94	0.94	0.94	0.94
Threonine, %	0.81	0.81	0.81	0.81	0.81	0.81	0.81	0.81	0.81	0.81
Tryptophan, %	0.26	0.26	0.26	0.26	0.26	0.26	0.26	0.26	0.26	0.26
Ca, %	1.00	1.00	1.00	1.00	1.00	1.00	1.00	1.00	1.00	1.00
P_T_, %	0.37	0.48	0.59	0.70	0.48	0.59	0.70	0.48	0.59	0.70
P_T_ (analyzed), %	0.36	0.50	0.60	0.72	0.48	0.59	0.68	0.50	0.59	0.71

^1^ Calcium carbonate (36% Ca); ^2^ Commercial calcium orthophosphate (21% P, 21% Ca); ^3^ NDP: nano-dicalcium phosphate (18% P, 24% Ca); ^4^ Analytical grade dicalcium phosphate (22% P, 30% Ca); ME = Metabolizable energy; P_a_ = Available phosphorus; P_T_ = Total phosphorus; ^5^ Mineral and vitamin premix containing: Zn, 100 g; Fe, 50 g; Cu, 10 g; Mn, 100 g; I, 1 g; retinol, 24,000,000 IU; cholecalciferol, 8,000,000 IU; pyridoxine, 8 g; thiamine, 6 g; riboflavin, 16 g; niacin, 100 g; cyanocobalamin, 60 mg; menadione, 10 g; calcium pantothenate, 28 g; folic acid, 3 g, as basis of 1000 g per ton of feed.

**Table 2 animals-11-02773-t002:** Growth performance in chicks fed with three sources and levels of supplementary phosphorous at 21 d of age.

P-Source/P_a_ (%)	Body Weight Gain (g)	Feed Intake (g)	Feed Conversion
Control			
0.13	228.40 ^d^	345.82 ^d^	1.55
EEM	19.17	33.62	0.05
Commercial			
0.24	600.50 ^c^	815.10 ^c^	1.35
0.35	756.80 ^a^	1001.69 ^abc^	1.31
0.46	799.20 ^a^	1073.40 ^a^	1.34
EEM	25.27	41.61	0.05
NDP			
0.24	616.90 ^bc^	862.34 ^bc^	1.40
0.35	745.20 ^ab^	1034.30 ^ab^	1.39
0.46	790.60 ^a^	1041.03 ^ab^	1.33
EEM	24.17	40.52	0.05
Analytical			
0.24	592.30 ^c^	812.3 ^c^	1.37
0.35	757.20 ^a^	1074.00 ^a^	1.42
0.46	748.40 ^ab^	972.49 ^abc^	1.30
EEM	23.27	36.62	0.05
Global			
Commercial	718.83	963.39	1.33
NDP	717.57	979.22	1.37
Analytical	659.30	952.93	1.36
EEM	21.18	26.45	0.06

Pa = Available phosphorous. EEM= Standard error mean; ^a–d^ Values in the same column with different superscripts were significantly different (*p* ≤ 0.05); P-Source Commercial: Tecamac Ultra Plus SA de CV; P-Source as nano-dicalcium phosphate (NDP): prepared particles in laboratory; P-Source Analytical: anhydrous dicalcium phosphate, J. T. Baker.

**Table 3 animals-11-02773-t003:** Consumption, digestibility and absorption of phosphorus (%) on day 10 and 21 in chicks fed with three sources and levels of supplementary phosphorous.

P-Source/P_a_ (%).	P_T_ Analyzed in Diet	Day 10				Day 21			
		T_PI_	T_PE_	T_PA_	A_DP_	T_PI_	T_PE_	T_PA_	A_DP_
Control									
0.13	0.36	0.68 ^d^	0.31^b^	0.35 ^c^	58.43 ^c^	0.57 ^d^	0.19 ^d^	0.38 ^c^	64.06 ^b^
EEM		0.07	0.08	0.07	5.29	0.21	0.22	0.19	4.32
Commercial									
0.24	0.50	1.17 ^c^	0.41 ^ab^	0.79 ^bc^	65.46 ^ad^	2.90 ^c^	0.87 ^bcd^	2.04 ^b^	70.17 ^a^
0.35	0.60	1.70 ^b^	0.70 ^ab^	0.99 ^ab^	59.79 ^c^	4.82 ^ab^	1.44 ^abc^	3.39 ^a^	70.32 ^a^
0.46	0.72	2.12 ^a^	0.77 ^a^	1.37 ^a^	64.01 ^ab^	5.66 ^a^	2.36 ^a^	3.30 ^a^	58.72 ^b^
EEM		0.06	0.07	0.08	4.89	0.20	0.19	0.18	4.12
NDP									
0.24	0.48	1.21 ^c^	0.49 ^ab^	0.75 ^bc^	61.39 ^a^	2.97 ^c^	0.91 ^bcd^	2.06 ^b^	69.25 ^a^
0.35	0.59	1.68 ^b^	0.54 ^ab^	0.99 ^ab^	66.93 ^bd^	4.56 ^b^	1.75 ^abc^	2.81 ^b^	61.52 ^b^
0.46	0.68	1.94 ^ab^	0.73 ^a^	1.16 ^ab^	62.31 ^a^	5.25 ^ab^	1.90 ^ab^	3.34 ^a^	63.86 ^b^
EEM		0.06	0.08	0.07	3.92	0.18	0.20	0.16	4.02
Analytical									
0.24	0.50	1.29 ^c^	0.55 ^ab^	0.72 ^bc^	55.65 ^c^	2.75 ^c^	0.81 ^cd^	1.94 ^b^	70.67 ^a^
0.35	0.59	1.67 ^b^	0.64 ^a^	0.84 ^b^	57.35 ^c^	4.69 ^ab^	1.78 ^abc^	2.91 ^b^	62.29 ^b^
0.46	0.71	1.92 ^ab^	1.00 ^a^	1.27 ^ab^	53.68 ^c^	4.68 ^ab^	1.39 ^abc^	3.29 ^b^	71.32 ^a^
EEM	0.04	0.04	0.06	0.06	4.19	0.21	0.18	0.16	3.92

T_PI_ = Total phosphorus intake; T_PE_ = Total phosphorus excreted; T_PA_ = Total phosphorus absorbed; A_DP_ = Apparent phosphorus digestibility; EEM = Standard error mean; P_a_= Available phosphorous; NDP= Nano dicalcium phosphate; ^a–d^ Values in the same column with different superscripts were significantly different (*p* ≤ 0.05); P-Source Commercial: Tecamac Ultra Plus SA de CV; P-Source NDP: prepared particles in laboratory; P-Source Analytical: anhydrous dicalcium phosphate, J. T. Baker.

**Table 4 animals-11-02773-t004:** Phosphorus and ash content in tissues collected after slaughter (21 days) of chicks fed with three sources and levels of supplementary phosphorous.

P-Source/P_a_ (%)	Breast P_T_ (mg/100g)	Liver P_T_ (mg/100g)	L_TB_ P_T_ (mg/100g)	L_TB_ Ash, %	R_TB_ T_L_ (mm)	R_TB_ T_PD_ (mm)	R_TB_ T_MD_ (mm)	R_TB_ T_DD_ (mm)
Control								
0.13	290 ^a^	360 ^a^	14980 ^b^	34.52 ^c^	52.27 ^c^	12.02 ^d^	4.27 ^c^	10.82 ^c^
EEM	6.0	8.0	700	0.97	1.36	0.59	0.28	0.41
Commercial								
0.24	310 ^b^	510 ^bc^	17430 ^ab^	44.03 ^b^	61.96 ^b^	16.19^c^	5.55 ^b^	13.96 ^b^
0.35	320 ^b^	390 ^a^	19450 ^a^	50.14 ^a^	69.39 ^a^	17.67^abc^	6.35 ^ab^	15.20 ^ab^
0.46	330 ^b^	430 ^b^	19450 ^a^	52.10 ^a^	69.86 ^a^	18.29^ab^	6.19 ^ab^	15.17 ^ab^
EEM	50	50	500	0.71	0.98	0.42	0.18	0.29
NDP								
0.24	310 ^b^	470 ^bc^	17210 ^ab^	45.95 ^b^	62.78 ^b^	16.46^abc^	5.60 ^ab^	14.25 ^ab^
0.35	390 ^c^	410 ^b^	17690 ^ab^	49.99 ^a^	69.27 ^a^	18.46^a^	6.18 ^ab^	15.40 ^a^
0.46	400 ^c^	350 ª	19120 ^a^	50.07 ^a^	69.46 ^a^	18.09^abc^	6.44 ^a^	14.90 ^ab^
EEM	50	50	600	0.71	1.00	0.43	0.18	0.30
Analytical								
0.24	320 ^b^	480 ^bc^	17320 ^ab^	44.31 ^b^	61.97 ^b^	16.30 ^bc^	5.69 ^ab^	14.35 ^ab^
0.35	300 ^b^	350 ª	18350 ^a^	50.60 ^a^	69.50 ^a^	17.92 ^abc^	6.26 ^ab^	15.46 ^a^
0.46	300 ^b^	530 ^c^	17360 ^ab^	53.21 ^a^	68.43 ^a^	17.74 ^abc^	5.99 ^ab^	14.67 ^ab^
EEM	50	53	600	0.75	1.04	0.45	0.19	0.31

PT =Total phosphorus; LTB = Left tibia bone; RTBTL = Total length of the right tibia; RTBTPD = Proximal diameter of the right tibia; RTBTMD = Medial diameter of the right tibia; RTBTDD = Distal diameter of the right tibia; EEM = Standard error mean; Pa = Available phosphorous; NDP = Nano dicalcium phosphate; ^a–d^ Values in the same column with different superscripts were significantly different (*p* ≤ 0.05); P-Source Commercial: Tecamac Ultra Plus SA de CV; P-Source NDP: prepared particles in laboratory; P-Source Analytical: anhydrous dicalcium phosphate, J. T. Baker.

## Data Availability

The data presented in this study are available on request from the corresponding author.

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
