# Peer review of "Designing Calcium Phosphate Nanoparticles with the Co-Precipitation Technique to Improve Phosphorous Availability in Broiler Chicks"

_animals, 2021, doi:10.3390/ani11102773_

Round 1

Reviewer 1 Report

The manuscript reports a study on calcium phosphate nanoparticles improving phosphorous availability in broiler chicks

Although the subject is attractive, the sample size is too small to have a valid conclusion. A limitation that should be addressed is the lack of detail in describing animal sampling as well statistical method. In more details:

Introduction

I would suggest to explain if the cost of NDP is much higher or lower than that of Commercial and analytical grade P sources

Materials and methods

In the Table 1, soghum content of o.24% commercial group is 8.21 ,but not 58.21?

Line 200 the author describe the analysis method of N content, but where is the results of N content?

Results

Please supplement the unit of dada in the table 3.

Author Response

I am attaching the file.

Reviewer 2 Report

The current study evaluated the effect of phosphate nanoparticle on the chicken performance and phosphorous digestibility. The data obtained could be helpful to the chicken industry. As the cost of the P has been emphasized by the authors as an expensive ingredient in chicken diet, the cost of supplementary P sources tested in the manuscript should also be discussed including the cost for fabricating the phosphate nanoparticles. In addition, several statements written in the results section are very confusing or too complicated to be understood by the readers. Authors are suggested to revise this section for better clarification and please avoid using too many “vs.” in the manuscript. Instead of using “vs.”, authors could use “compare”. One of the main objectives of the manuscript was to determine the effect of different P sources especially the use of phosphate nanoparticle on the chicken performance and P digestibility. Therefore, authors are suggested to focus more on the individual P source results (especially the results of nanoparticle group as compared to the control and the positive control) in the result section, other than just use “supplementary P” as general statements. Specific comments are listed below.

Line 32, what was the P source in the control diet?

Line 33-34, please provide which P source (commercial, DP, or NDP) at which P level showed the greatest performance.

Line 34, Remove “vs. control” and use “as compared to the control”.   

Line 37, Were all kinds of supplemental P sources at all concentrations significantly higher than the control? Again, please remove “vs. control” and use “as compared to the control”

Line 57, please define bigger size

Introduction is too short. More contents should be provided such as chicken P digestibility and the design of NDP

Line 85, CaCl2 (lower subscript)

Line 95, how was the concentration of the phosphorus calculated as in the nanoparticle format?

Line 100-102, “Subsequently, …….nanoparticles formed.” Please rephrase the sentence for better clarification.

Line 119, was the stability of the nanoparticle analyzed?

Line 160, provide sources for these P sources and the information on how were the treatment diets prepared. How was the NDP used? By dissolving in water or directly used as in powder format? Was the stability of the NDP measured since the diets were prepared weekly not daily.

Line 162, what were the 4 x 10 factors, please define  

Line 168, table 1, 24, 35, 46% or 0.24, 0.35, and 0.46% as indicated in Line 162? Please clarify. And please also define the abbreviation “P” in the manuscript. For example, control diet P=0.13% (Line 160), is this P referring as available P or total P?   

Table 1, Commercial, 24% Pa, L-Triptophane “02”? Is “02” the correct value to be reported? Please check   

Table 1, Commercial, 25% Pa, Sorghum at 8.21 as compared to other diets as ~ 58, why such huge difference? And provide units for the ingredients.

Line 178, replace “by” by “per”

Line 255, since the source of P was also one of the important factors studied, chicken performance results of each P source at different P concentrations should be briefly explained in this paragraph.

Table 3, the statistical analysis didn’t seem right in this table. For example, the Adp on day 10, 0.24 Commercial at 65.46 was not significantly different from 61.39 (NDP, 0.24) but significantly differed from 66.93 (NDP, 0.35)? Please recheck the SAS output to ensure the presented results were correct throughout the entire table.

Line 272, Was the consumption of P significantly higher at 21 d than that of 10d? Please clarify.

Line 273, use “among” but not “between”

Line 290, regardless the source of the P?

Line 306-308, please provide citations.

Line 344, significantly as compared to the control group?

Author Response

I am attaching the file

Round 2

Reviewer 1 Report

Accept in present form.